# A Novel RHS1 Locus in Rice Attributes Seed-Pod Shattering by the Regulation of Endogenous S-Nitrosothiols

**DOI:** 10.3390/ijms232113225

**Published:** 2022-10-30

**Authors:** Bong-Gyu Mun, Muhammad Shahid, Gang Sub Lee, Adil Hussain, Byung-Wook Yun

**Affiliations:** 1School of Applied Bioscience, College of Agriculture and Life Science, Kyungpook National University, Daegu 41566, Korea; 2Agriculture Research Institute, Khyber Pakhtunkhwa, Mingora 19130, Pakistan; 3Biosafety Division, National Institute of Agricultural Science, Jeonju, 54875, Korea; 4Department of Agriculture, Abdul Wali Khan University, Mardan 23200, Pakistan

**Keywords:** seed shattering, rice, nitric oxide

## Abstract

Seed or pod shattering in rice (*Oryza* sativa) is considered to be one of the major factors involved in the domestication of rice as a crop. High seed shattering results in significant yield losses. In this study, we characterize the *RICE*
*HIGH*
*SHATTERING 1* (*RHS1*) that corresponds to the locus *LOC_Os04g41250* from a greenhouse screen, involving 145 Ac/Ds transposon mutant rice lines. The knockout mutant line *rhs1* exhibited a significantly high shattering of grains in comparison to the wild-type plants. The exogenous application of nitric oxide (NO) resulted in a significant reduction in the expression of *RHS1* in wild-type rice plants. The absence of *RHS1*, which encodes a putative armadillo/beta-catenin repeat family protein, resulted in high sensitivity of the *rhs1* plants to nitrosative stress. Interestingly, the basal expression levels of *QSH1* and *SHAT1* genes (transcription factors that regulate seed-pod shattering in rice) were significantly lower in these plants than in wild-type plants; however, nitrosative stress negatively regulated the expression of *QSH1* and *SHAT1* in both WT and *rhs1* plants, but positively regulated *QSH4* expression in *rhs1* plants alone. The expression levels of genes responsible for NO production (*OsNIA1*, *OsNIA2,* and *OsNOA1*) were lower in *rhs1* plants than in WT plants under normal conditions. However, under nitrosative stress, the expression of *OsNIA2* significantly increased in *rhs1* plants. The expression of *CPL1* (a negative regulator of seed shattering in rice) was significantly lower in *rhs1* plants, and we found that *CPL1* expression was correlated with *S*-nitrosothiol (SNO) alteration in *rhs1.* Interestingly *noe1*, a rice mutant with high SNO levels, exhibited low seed shattering, whereas *rhs1* resulted in low SNO levels with high seed shattering. Therefore, *RHS1* is a novel gene that negatively regulates the shattering trait in rice via regulation of endogenous SNO levels. However, the molecular mechanisms involved in the control of *RHS1*-mediated regulation of seed shattering and its interaction with nitric oxide and involvement in plant defense need to be investigated further.

## 1. Introduction

Rice (*Oryza sativa*), one of the major crops used as staple food worldwide, is consumed by more than 50% of the world’s population and is the most rapidly growing cereal source to overcome food scarcity after wheat [1]. A number of traits, such as water requirements, early maturity, growth speed and habits, plant height, stem strength, seed size and nutrient composition, starch and protein content in grains, and various other properties related to plant architecture, play a key role in the domestication of rice in Asia and Africa: this has led to the adaptation of two major types of rice species, i.e., *Oryza sativa* and *Oryza glaberrima*, respectively [2]. In addition to these, seed shattering is considered one of the major factors affecting the domestication of rice [3] via natural and/or artificial selection processes throughout history. The shattering trait of wild rice severely reduces its yield. However, upon maturity, if the seeds remain attached to the panicle for a longer duration, or until harvesting and threshing, the yield will increase significantly [4,5]. Seed shattering thresholds are also considered important in the design of harvesting and threshing machines in order to minimize yield loss. For example, hard or non-shattering cultivars should be harvested and threshed using small head-feeding combine harvesters, whereas moderate shattering cultivars should be harvested and threshed using large combine harvesting machines [6].

Various studies on rice seed shattering have been conducted using molecular markers and by considering seed shattering a quantitative trait controlled by unknown genes [7]. However, Konishi et al. [8] showed that the loss of a function mutation of the quantitative trait locus (QTL) in chromosome 1 (*qSH1*), which encodes a BEL1-type homeobox protein, exhibits a loss of seed shattering by inhibiting the formation of an abscission layer at the base of the grains. This was caused by a single nucleotide polymorphism (SNP) in the 5′ upstream regulatory region of *qSH1,* which resulted in the loss of its expression in the abscission layer, decreasing seed shattering and yielding rice domestication.

Similarly, *SH4*, which encodes a trihelix family transcription factor, has been shown to promote the hydrolysis of abscission zone (AZ) cells during the abscission process [3]. Subsequently, the seed shattering abortion 1 (*SHAT1*), which encodes an APETALA2 transcription factor, was shown to be required for seed shattering in rice because it controls the development of the AZ in spikelets, and its expression is positively regulated by *SH4* [9]. The AP2 transcription factors are highly conserved and widespread across the plant kingdom and are involved in diverse physiological processes, such as metabolic pathways, hormone biosynthesis, and biotic and abiotic stresses [10]. Thus, these transcription factors have been considered valuable targets for genetic manipulation in order to incorporate desired traits in crop plants via genetic engineering [11]. At least three different quantitative trait loci (QTLs)—sh3, sh4, and sh8—responsible for the reduction in seed shattering were identified through genetic analysis of an F2 population derived from a cross between *O. sativa* and *O. nivara* [12]. Ji et al. [13], in 2010, described the rice sh-h gene LOC_Os07g10690. RNAi lines with a suppressed expression of this gene exhibited greater shattering. The gene encodes a protein containing a conserved carboxy-terminal domain (CTD) phosphatase-like domain (OsCPL1). Their results regarding subcellular localization and biochemical analysis revealed that the OsCPL1 is a nuclear phosphatase, a common characteristic of metazoan CTD phosphatases involved in cell differentiation. They showed that OsCPL1 represses the differentiation of the abscission layer during panicle development in rice, thereby negatively regulating seed shattering.

The present species of rice grown globally originated from one, or perhaps both, of the closely related wild species, *O. rufipogon* and *O. nivara*, which can be found throughout South East Asia [14,15,16]. One interesting fact about seed shattering is the color of the seed hull. Green immature seed hulls of cultivated rice (*O. sativa*) turn straw-white upon maturity. However, seed hulls of progenitor wild rice cultivars tend to turn black, darkening at maturity. The darker color of seed hulls is always associated with high seed shattering in rice [17]. The black hull color has been hypothesized to serve as a camouflage for the shattered seeds in the soil. Although the black hull color is known to be controlled by *black hull* (*BH*) genes, the molecular mechanisms underlying the change in seed color from green to black or straw-white remain unknown.

Various reactive oxygen species (ROS) and reactive nitrogen species (RNS) comprise a group of redox signaling molecules that regulate the inter- and intra-cellular networks. Both ROS and RNS are small molecules that play a role in altering the cellular redox state to activate and/or deactivate specific physiological pathways, both under normal conditions and during plant responses to stress [18,19]. Nitric oxide (NO) signaling is mediated by various NO derivatives, such as NO radical (∙NO), nitroxyl (NO^−^), peroxynitrite (ONOO^−^), and S-nitrosothiols (SNO) [20].

NO is implicated in the control of seed dormancy and germination [21], although the details of the exact molecular mechanisms are yet not known. Various nitrogen-containing compounds influence seed dormancy and germination [22]. Assuming that all these molecules function in a similar way, NO might be a possible candidate to understand these characteristics. Because seed dormancy is regulated by different phytohormones, the role of NO in modulating the levels of different phytohormones under normal and stress conditions is well known. Numerous studies have explored the role of NO in mediating different physiological processes in plants. However, the relationship of NO with seed shattering is not yet known. In this study, we report an association between NO signaling and seed shattering in rice.

## 2. Methods

### 2.1. Plant Materials and Phenotype Evaluation

A population of 145 Ac/Ds transposon mutant rice lines from a *Dongjin* background was obtained from the Rural Development Administration, Korea. Initial screening revealed that the loss of function in transposon mutant lines of rice *Loc_Os04g41250*, which encodes a putative armadillo/beta-catenin repeat family protein, exhibited high seed shattering, and was therefore named as rice high shattering 1 (RHS1). Plants were grown in a paddy at a research field at the Kyungpook National University, Gunwi, Republic of Korea. Two locally cultivated rice cultivars, “Woonkwang” and “Drimi”, were also cultivated in the field as the controls. Normal agricultural practices were adopted to grow the cultivars. Data on phenotypic characteristics, such as the average number of tillers per plant, length of culm and panicle, and number of spikelets per panicle, were recorded for each line. The response to nitro-oxidative stress was analyzed by sterilizing rice seeds with 25% prochloraz and germinating them on sterile wet tissue papers for 12 days at 24 °C. The plants were then transferred to 13 cm × 9 cm 6-well tissue culture test plates (SPL, Pocheon, Korea), containing S-nitrosocysteine (CysNO), for 6, 12, and 24 h at 24 °C. Data on the number of tillers/plants, number of spikelets/spikes, culm length, and panicle length were recorded.

### 2.2. Nitro-Oxidative Stress Treatment

A CysNO solution was prepared by mixing equal volumes of 1 M l-cysteine dissolved in 1 N HCl and 1 M sodium nitrate (NaNO_2_). The resulting solution was diluted with 40 mM N-(2-hydroxyethyl) piperazine-N′-(3-propane sulfonic acid) (pH 7.8) to yield 10 mM CysNO. Thus, 2-week-old MS-grown plants were treated with 10 mM and 50 mM CysNO in 6-well plates for 6, 12, and 24 h. The plates were maintained in the dark to prevent the decomposition of CysNO.

### 2.3. Measurement of Seed Shattering

The rice-seed-shattering degree was measured as described by Qin, Kim, Zhao, Jia, Lee, Kim, Eun, Jin, and Sohn [5]. Briefly, the primary tillers of rice plants were harvested 45 days after the start of heading and maintained at room temperature for a week, or until the moisture of the spikelets reached approximately 20% to 25%. Next, 20 grains at three different points on each panicle were analyzed for their breaking tensile strength and the pulling strength required to shear them off their pedicels. For this, the pedicle of each grain was fixed upside down to a digital force gauge (Model No. DS2-5N; IMADA, Tokyo, Japan), and the grains were pulled down using forceps. The value obtained using the digital force gauge was recorded at the moment of pedicle breakage in gravitational force unit (gf).

### 2.4. SNO Measurement

To measure SNO contents, rice plants were grown for four weeks with the Hoagland solution in a Magenta box, and then five plants from each box were harvested. Samples were ground with liquid nitrogen until a fine powder was obtained. Then, 1ml of extraction buffer (1× PBS pH 7.4) was added to the powdered samples and mixed well. The samples in PBS were centrifuged at 13,500× *g* for 10 min at 4 °C. The supernatant was transferred and centrifuged again at 15,900× *g* for 10 min at 4 °C. Proteins were quantified by the Bradford assay method using a Coomassie (Bradford) protein assay kit (Thermo Fisher Scientific, Waltham, MA, USA), according to the manufacturer’s manual. Briefly, 1.5 mL of Coomassie dye reagent was added to 30 µL of extracted protein and mixed thoroughly. The optical density (OD) of all the samples were measured with a spectrophotometer (OPTIZEN α, Mecasys, Daejeon, Korea) at 595 nm. For the SNO measurement, 100 µL of the extracted proteins were injected into the reaction vessel of the Nitric Oxide Analyzer (NOA 280i, Sievers, Estero, FL, USA) containing a CuCl/cysteine reducing agent, and the peak values were recorded. The SNO content was derived from a standard curve generated by CySNO standards. As a final step, the SNO level was calculated as nM/µg protein.

### 2.5. RNA Extraction and Real-Time Polymerase Chain Reaction

RNA was extracted from the control and the treated plants by using Trizol^®®^ (Invitrogen, Carlsbad, CA, USA), according to the manufacturer’s instructions. The cDNA was synthesized using a cDNA synthesis kit (PhileKorea, Seoul, Korea), according to the manufacturer’s instructions. Quantitative real-time polymerase chain reaction (qRT-PCR) was performed using an Eco^TM^ real-time PCR machine (Illumina, San Diego, CA, USA), using a 2× Quantispeed SYBR Mix (PhileKorea, Korea), along with 100 ng of template DNA and 10 nM of each primer, in a final volume of 20 µL in a two-step PCR for 40 cycles under the following conditions: polymerase activation at 95 °C for 2 min; subsequent denaturation steps at 95 °C for 5 s; and annealing and extension at 65 °C for 30 s. Rice ubiquitin and a “no template control” were used as an internal reference and negative controls, respectively. Primer sequences are shown in Appendix A.

### 2.6. Statistical Analysis

Each experiment was performed at least three times with multiple replicates. Means were analyzed for significant difference using the Student’s *t*-test.

## 3. Results

### 3.1. Screening the Ac/Ds Transposon Mutant Rice Lines

While growing 145 Ac/Ds transposon mutant rice lines from a *Dongjin* background over a period of two years, we found interesting phenotypic characteristics, such as being dwarf-sized, being taller, having a lesser number of tillers, and having early/or late maturing lines (Appendix A). For further detailed analysis, we selected eight mutant lines that exhibited interesting phenotypes and looked for their gene names and annotation (Appendix A). The line with the loss of the LOC_Os04g41250 function exhibited significantly higher seed-pod shattering as we could see the seed pods on the ground. LOC_Os04g41250 encodes a putative armadillo/beta-catenin repeat family protein known to be involved in cadherin-associated cell adhesion to the cytoskeleton in *Drosophila*, and it is involved in a broad range of biological functions. In addition, we also recorded significant changes in the expression of this gene in the transcriptomic analysis performed after the external application of CySNO in both *Arabidopsis* [23] and rice [24]. The protein sequence and its predicted 3D structure were found to be significantly conserved and similar to the armadillo domain of beta-catenin protein XP_033152121 from *Drosophila* (Appendix A), which is an key protein involved in Wnt/Wingless signaling in *Drosophila* [25]. We further measured seed-pod shattering by measuring the horizontal, as well as the vertical, force required to mechanically detach the seed pod from the plant (represented as “force value”).

### 3.2. Loss of RHS1 Function Leads to Seed Shattering and Increases Sensitivity to Nitrosative Stress

Analysis of *rsh1* plants showed that the seeds of these plants needed 35% and 50% less vertical and horizontal force, respectively, to pluck the grain from inside the spikelet, compared to that required for the wild-type (WT) plants and those of the other two cultivars, Wonkwang and Drimi-2ho (Figure 1A). Nitric oxide has been widely recognized as a master regulator of a plethora of physiological processes in plants. However, the role of NO in regulating seed shattering is not yet known. In a previous study, in the transcriptomic analysis of rice induced by 1 mM CysNO, it was revealed that RHS1 was down-regulated (Appendix A). Accordingly, NO was applied exogenously to investigate any linkage among NO, RHS1, and shattering-related known genes. As a result, the expression of shattering-related genes, especially SHAT1, exhibited a significantly lower expression in the *rhs1* plants compared to the WT plants under normal conditions. Apart from that, nitrosative stress negatively regulated the expression of QSH1 and SHAT1 in both the WT and *rhs1* plants, but it positively regulated QSH4 expression in only the *rhs1* plants (Figure 1B).

Furthermore, many studies have shown that plants with higher levels of S-nitrosothiols exhibit a stunted phenotype, perturbed pollination, loss of viability or lower germination frequency, sterility, smaller leaves, and various abnormal floral and seed phenotypes [26]. Similarly, the *rsh1* plants exhibited a relatively stunted phenotype (Appendix A) and produced fewer seeds per panicle (subfigrue C in Section 3.4); thus, we attempted to evaluate the sensitivity of *rsh1* mutants to nitrosative stress. For this, the WT and *rsh1* plants were treated with 10 and 50 mM CySNO. The *rhs1* seedlings showed an overall poor growth and exhibited acute sensitivity to nitrosative stress. Their leaves turned yellow within 12 h of treatment and their leaf tips became dried, necrotic, and turned brown, eventually leading to the death of the plants within 24 h (Figure 2A). Additionally, nitrosative stress caused a reduction in biomass plants (Appendix A). On the other hand, the WT plants showed less symptoms of nitrosative stress. This indicates that the absence of the *RHS1* function makes plants sensitive to nitrosative stress, either due to a higher accumulation of NO at the cellular level or its associated forms, or due to loss of NO-scavenging ability. Quantitative real-time PCR analysis of nitric-oxide-related genes showed that, in *rhs1* plants, the expression levels of genes responsible for NO production (*OsNIA1*, *OsNIA2,* and *OsNOA1*) were lower compared to that of *OsGSNOR* under normal conditions (Figure 2B–E). However, after treatment with the nitric oxide donor, we found a gradual increase in the expression of *OsNIA2*, which was responsible for the maximum amount of nitrate reductase activity or NO production through the reductive pathway (Figure 3E). Although *OsNIA1* (Figure 2B) and *OsNOA1* (Figure 2C) showed slight variations in their expression following nitrosative stress, 10 mM CySNO proved to be a terminal dose for a significant increase in their expression at 48 h post-treatment. These results were further supported by the lower expression of *OsGSNOR* in the *rhs1* mutants (Figure 2D).

### 3.3. High Seed Shattering in rhs1 Is Associated with Cellular SNO Content

The higher relative expression of *OsGSNOR*, and the lower expression of NO production genes in the *rhs1* plants, under normal conditions indicated a possible involvement of nitric oxide, or its derivatives, in seed shattering. Our results revealed a significantly higher force value for the *noe1* (*nitric oxide excess 1*) mutant compared to the WT and *rhs1*, indicating that *noe1* needs a higher force, both vertically and horizontally, to pluck the seeds from the spikelet (Figure 3A). Therefore, in order to investigate the possible role of endogenous SNO content and seed shattering, we measured the total SNO content of the *rhs1*, WT (Dongjin and Nipponbare), and the NO-overproducing mutant *noe1* plants. The results showed significantly lower SNO levels in the *rhs1* mutant lines, while *noe1* exhibited significantly higher SNO contents compared to the wild-type and *rhs1* (Figure 3B), indicating that the alteration of SNO content affects seed shattering. This result was further supported by the significantly lower expression of *CPL1* (a negative regulator of seed shattering which represses the development of the abscission layer required for seed shattering) in *rhs1* plants and the significantly higher expression in *noe1* plants (Figure 3C). Altogether, our findings reveal that the loss of *RHS1* function improves seed shattering in rice where endogenous SNO contents of *rhs1* plants may play an important role, directly or indirectly.

### 3.4. RHS1 Is Required for Normal Growth and Development in Rice

*RHS1* is a member of the zinc finger family of genes and encodes an armadillo/beta catenin repeat family protein. The gene consists of 5′ and 3′ UTR and 4 introns between the translational start and stop codons (Appendix A). Quantitative real-time PCR analysis showed that *RHS1* is mainly expressed in seeds with at least a 1000-fold-high expression compared to the leaves and stems. The highest expression was recorded in mature seeds, followed by seeds in the milking stage, and then roots. (Figure 4A and 4A inset). Rice *rhs1* plants were relatively shorter in height (Appendix A). The loss of the *RHS1* function also resulted in a significant reduction in the number of tillers compared to the WT plants (Figure 4B) and spikelets per spike (Figure 4C), while these plants also produced shorter culms and panicles (Figure 4D).

## 4. Discussion

Seed shattering is the natural detachment and dispersal of seeds upon reaching physiological maturity. Scientific studies in the literature suggest that the loss of shattering in crops is the result of genetic variations or mutations occurring over a long period of time, due to which plants tend to retain the seeds after physiological maturity, hence making harvesting possible [27]. The genetic control of seed shattering has been investigated in all major crops, such as barley [28,29], rice [3,6,8,30], millet [31], buckwheat [32], wheat [33,34,35,36], maize [37], rye [38], sorghum [39], and many others. Among them, most of the genes are affiliated with transcription factors.

Nitric oxide is a free radical and acts as a small signaling molecule. A number of studies with exogenous NO application demonstrated that it affects regulation of genes involved in diverse physiological pathways, such as redox signaling, hormone signaling, various protein kinases, and transcription factors [23,40]. Germination and pollen development are also known to regulate seed dormancy by controlling hormone signaling, along with other features of plant reproductive structures [41]. However, information about NO-mediated seed shattering is not available so far. To investigate the link between nitric oxide and seed shattering, we screened 145 Ac/Ds transposon mutant rice lines from a *Dongjin* background for their phenotypical characteristics, where we found a rice mutant exhibiting abnormal growth and a high seed-pod shattering trait (Figure 1A). *rhs1* encodes a putative armadillo/beta-catenin repeat family protein, and it was found in our previous transcriptomic data to be down-regulated by almost two-fold (Appendix A). In addition, according to its annotation, it is known to be involved in cadherin-associated cell adhesion to the cytoskeleton in *Drosophila*, and these proteins are involved in a broad range of biological functions. Recently, some genes of the ARM repeat family have been reported to be involved in GA signaling and ABA-dependent gene expression in plants [42,43]. Moreover, based on protein sequence similarity, *RHS1* orthologues revealed their presence in *Arabidopsis*, maize, and other plant species. Depending on the gene expression results shown in Figure 4A, *rhs1* is highly expressed in the seeds compared to the stem, leaves, and roots. Interestingly, shattering-related genes, such as *qSH1* (LOC_Os01g62920), *SH4* (LOC_Os04g57530), and *SHAT1* (LOC_Os04g55560), also follow the same expression patterns with maximum RNA-Seq FPKM expression values in the inflorescence for *qSH1* and *SH4* and in the endosperm for *SHAT1* (http://rice.plantbiology.msu.edu/index.shtml: accessed on 1 March 2022). However, the unique function of *RHS1* is that it negatively regulates seed shattering in contrast to the other genes mentioned above that positively regulate seed shattering, as their loss-of-function mutant plants exhibited non-shattering phenotypes.

The *RHS1* locus identified in this study not only regulates seed shattering, but also affects other growth parameters (Figure 4C,D). In general, plants with an impaired redox balance or the loss-of-function of essential gene(s) show abnormal growth and phenotypes. With ROS, nitric oxide is an important redox-active molecule which plays an important role in various physiological processes, as has been revealed by numerous studies involving plants accumulating high or low levels of NO in *Arabidopsis*. The literature indicates that plants with a stunted phenotype (Appendix A), perturbed pollination, and various disturbed floral and seed phenotypes are caused due to changes in endogenous NO [26]. In rice, Lin et al. reported that *noe1* generates more NO in plants exhibiting PCD (programed cell death)-like phenotypes. Taken together, we expected that the abnormal phenotype and high seed shattering of *rhs1* could be caused by either a change in the NO contents or NO-related signaling pathway. Therefore, we investigated the response of *rhs1* plants to nitrosative stress. As shown in Figure 1B, the expression levels of the two transcription factor genes, *QSH1* and *SHAT1*, were down-regulated in the presence of NO in both the WT and *rhs1*. However, the absence of *rhs1* resulted in an increased expression of *QSH4*. In addition, a high dose of nitrosative stress to *rhs1* plants resulted in a higher sensitivity. Three quantitative trait loci (QTLs) have been actively studied in relation to seed shattering in rice. These are qSH1, qSH3, and qSH4 [30]. Significantly different patterns of genealogical relationships have been found in the two major QTLs, qSH1 and qSH4, in regulating seed shattering. For example, Onishi, Takagi, Kontani, Tanaka, and Sano [30] stated that qSH1 is genetically epistatic in comparison to the other two loci. Furthermore, it has been reported that a single nucleotide substitution of G to T in qsh4 is responsible for reduced seed shattering in rice cultivar Nipponbare [8]. This indicates a significantly important, yet unique, role of qSH4 in regulating seed shattering compared to the other loci. As to why NO positively regulates the expression of qSH4 in only rhs1 plants is not clear and needs further investigation. Nonetheless, these results may suggest that NO, as one of the master signaling molecules, possibly mediates the regulation of shattering-related genes and *rhs1* may associate with NO, either directly or indirectly. Furthermore, *rhs1* seedlings exhibited a gradual increase in the expression of *OsNIA1* and *OsNIA2* in response to NO (Figure 3). Our results also revealed a higher expression of *OsGSNOR* and a lower expression of NO-production-associated genes in *rhs1* compared to the WT under normal conditions. We found that the absence of *RHS1* leads to the failure of adjustment in response to exogenous NO application. We further found that the loss of the *RHS1* function leads to a reduction in endogenous SNO contents. Upon demonstration, it was revealed that the high SNO-containing rice mutant, *noe1*, exhibited a higher force value, while *rhs1* had a lower SNO content and a lower force value. These results indicate a possible involvement of endogenous SNO contents in seed shattering. Moreover, the expression of *OsCPL1*, the negative regulator of seed shattering by repressing abscission layer differentiation in the process of shattering, was lower in the loss of function of *rhs1* while it was significantly higher in the NO over accumulator *noe1*, compared to both of the wild-type rice plants. Altogether, these results indicate that *RHS1* negatively regulates seed shattering and that higher endogenous SNO levels may add to lower seed shattering in rice.

This study reports the involvement of a novel gene, *RHS1*, in seed shattering. As mentioned earlier, most of the shattering-related genes or QTLs reported so far in the literature positively regulate seed shattering as their knockout or loss-of-function mutants exhibit non-shattering phenotypes. Meanwhile, *RHS1* negatively regulates shattering as the loss of function of *rhs1* plants exhibits high seed shattering with perturbed SNO contents. The molecular and mechanistic control of the *RHS1*-mediated regulation of seed shattering and its interaction with nitric oxide needs to be investigated further.

## Figures and Tables

**Figure 1 ijms-23-13225-f001:**
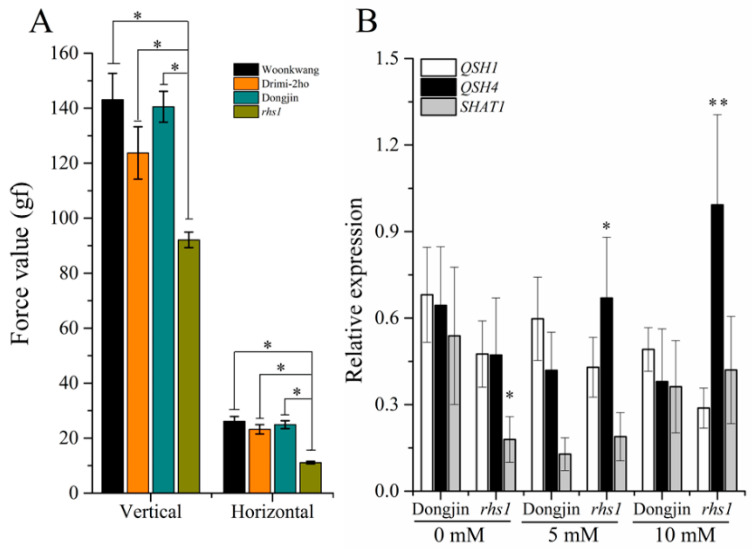
*RHS1* regulates seed-pod shattering in rice. The seeds of the *rhs1* plants showed significantly lower force values in terms of the vertical and horizontal force required to mechanically detach them from the base of the spikelet (**A**). A significant variation was observed in the expression of shattering-related genes in the *rhs1* plants under normal and nitrosative stress conditions (**B**). Each data point represents the average of three data points. Means were analyzed for significant difference using the Student’s t-test. Error bars represent the standard deviation. Significant differences are represented by asterisk (*) and (**) at *p* < 0.05 and *p* < 0.01, respectively.

**Figure 2 ijms-23-13225-f002:**
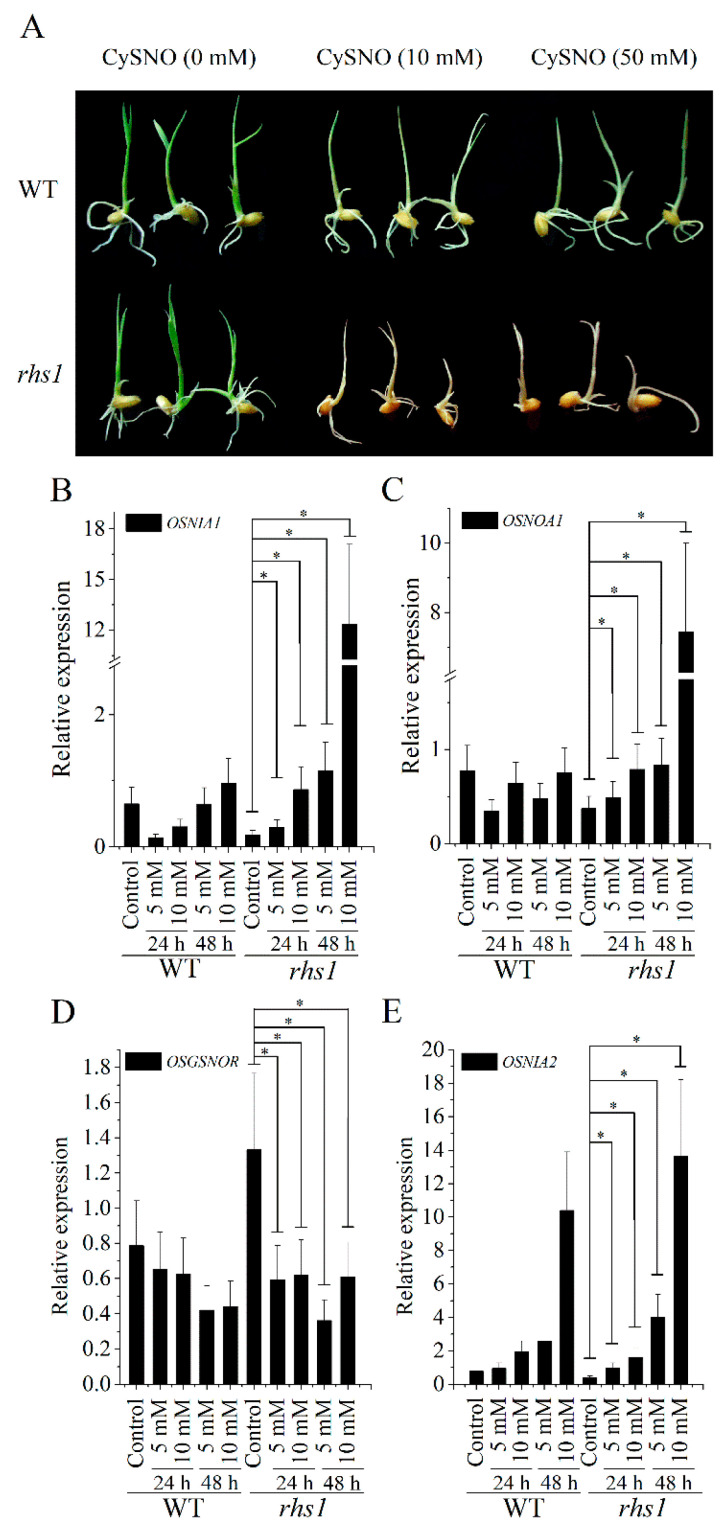
Response of *rhs1* seedlings to nitrosative stress. The seedlings of *rhs1* tested for their response to nitrosative stress were induced by 10 and 50 mM CysNO (**A**). Expression of rice *NIA1* (**B**), *NOA1* (**C**), *GSNOR* (**D**), and *NIA2* (**E**) before and after nitrosative stress treatment is shown. Each data point represents the mean of the three replicates. Means were analyzed for significant difference using the Student’s *t*-test. Error bars represent the standard deviation. Significant differences are represented by an asterisk (*) at *p* < 0.05.

**Figure 3 ijms-23-13225-f003:**
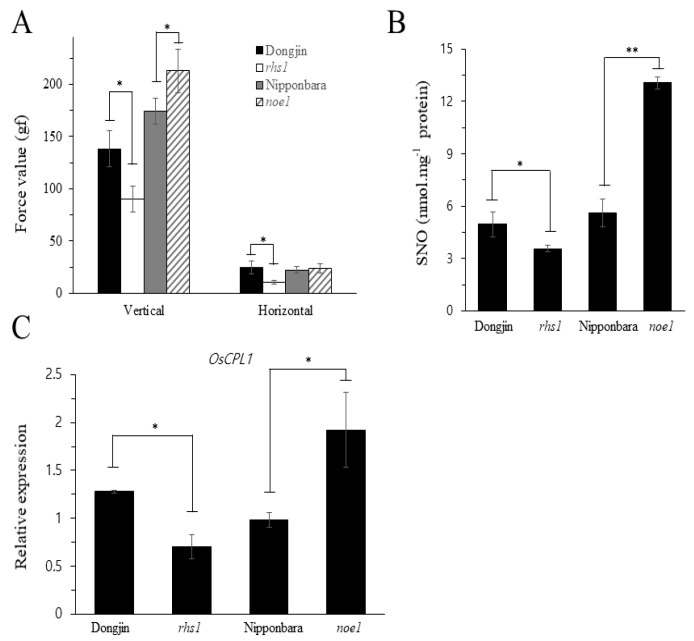
*RHS1* regulates seed shattering and SNO contents in rice. The seeds of *rhs1* plants showed a significantly lower force value in terms of the vertical and horizontal force required to pluck them from the base of the spikelet (**A**). The endogenous level of SNO was significantly lower in the *rhs1* plants compared to the WT, and a high SNO-containing mutant (*noe1*) exhibited higher SNO levels (**B**). The expression of rice *CPL1*, the negative regulator of seed shattering, is shown (**C**). Each data point represents the average of three data points. Means were analyzed for significant difference using the Student’s *t*-test. Error bars represent the standard deviation. Significant differences are represented by an asterisk (*) at *p* < 0.05, (**) *p* < 0.01.

**Figure 4 ijms-23-13225-f004:**
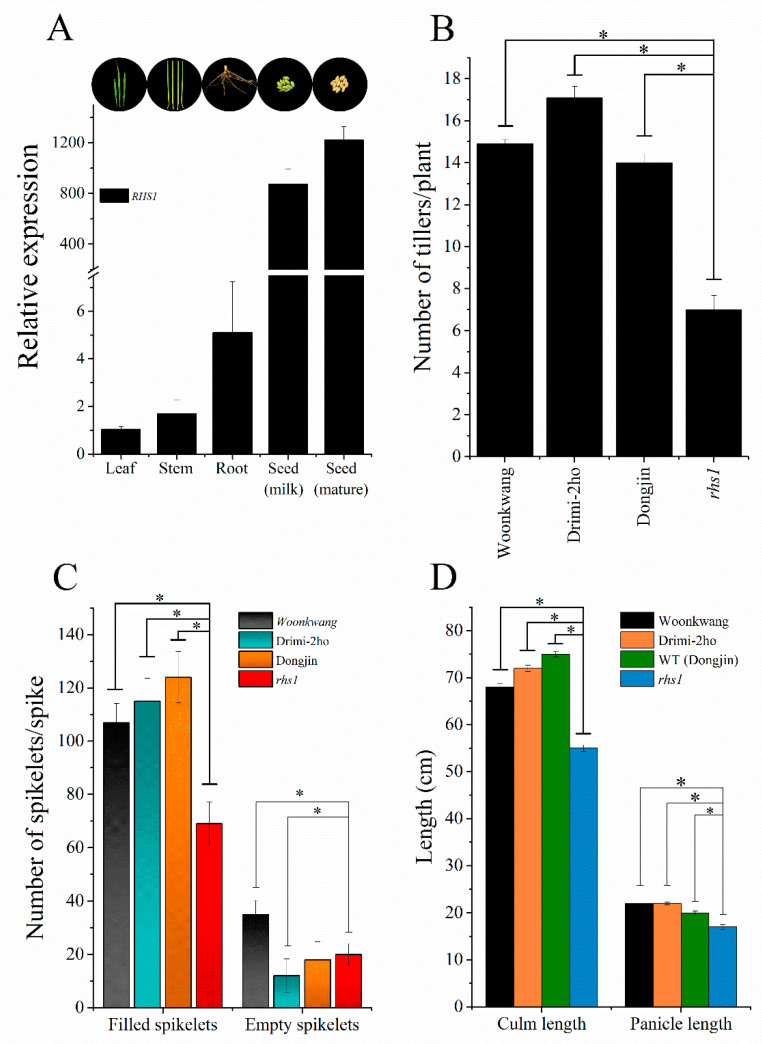
*RHS1* is expressed in seeds and regulates plant development. The expression of *RHS1* (LOC_Os04g41250) was highest in mature seeds, followed by milk-stage seeds and roots (**A**). The loss of the *RHS1* function negatively affected the overall yield and plant development as the *rhs1* plants produced a smaller number of tillers per plant (**B**), a smaller number of spikelets per spike (**C**), and showed a stunted phenotype (**D**). Each data point represents the mean of three replications. Means were analyzed for significant difference using the Student’s *t*-test. Error bars represent the standard deviation. Significant differences are represented by asterisk (*) at *p <* 0.05.

## Data Availability

Not applicable.

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
