# Peer review of "A Novel RHS1 Locus in Rice Attributes Seed-Pod Shattering by the Regulation of Endogenous S-Nitrosothiols"

_ijms, 2022, doi:10.3390/ijms232113225_

Round 1
Reviewer 1 Report
Seed or pod shattering in rice (Oryza sativa) is considered one of the major factors affecting the domestication of rice, and directly or indirectly affect its yield. In this study, the authors characterized a novel gene RHS1 that negatively regulates shattering via regulation of endogenous SNO levels. While, this story is a little bit simple and results were made mainly based on transcriptional level analysis. In order to make the story more reliable and complete, the molecular mechanisms involved in the control of RHS1-mediated regulation of seed shattering should be appropriately supplemented. Until these issues are resolved, I think that it can’t be accepted on “International Journal of Molecular Sciences” in current form. Three of my main concerns and uncertainties are listed below.
Major concerns:
1. Lines 133-134, “NO was applied exogenously to investigate any linkage among NO, RHS1 and shattering related known genes…” Lines 157-160, “Quantitative real time PCR analysis of nitric oxide related genes showed that in rhs1 plants…” In view of the fact that many studies related to NO have been mentioned in the text, relevant literature research progress should be appropriately added in the introduction.
2. Lines 136-138, “Nitrosative stress negatively regulated the expression of QSH1 and SHAT1 in both WT and rhs1 plants but positively regulated QSH4 expression in rhs1 plants only…” Did the authors attempt to explain the main reason for positive regulation of QSH4 expression only in rhs1 plants? If so, it will be of great help to study the relationship of NO with seed shattering.
3. There are problems with the saliency marks in many figures, such as the saliency marks in Figure 1B, Figure 3C and so on.
Author Response
Thanks for the valuable comments. We addressed all the comment diligently

Reviewer 2 Report
1. The title should be rewritten in a correct English grammar form.
2. Lines 125 – 129, it would be better if the authors can mention why did they choose and include two other rice cultivars in the experiment shown in Fig. 1a. Then the readers can understand the experimental design for other Fig. 3 and 4 (also including other two rice cultivars) as well.
3. I suggest re-arranging Fig. 1b to be 3 graphs showing the genes separately. The authors should statistically compare the gene transcript levels between WT and rhs1 mutant.
4. For all Figs, please mention the p-value for the statistical analysis (*).
5. Please check the whole manuscript and make sure the gene names should be written in italic form.
6. Please first describe all genes' full names with abbreviated names in the parentheses. Then the short names can be used afterward.
Author Response

(The authors gave the same response as above.)

Reviewer 3 Report
The authors report of the identification of RHS1 that encodes a putative armadillo/beta-catenin repeat family protein affecting seed-shattering in rice. RHS1 is mainly expressed in mature seeds and reduces the expression of some nitric oxide-related genes, such as OsNIA1, OsNIA2, and OsNOA1 in the mutant rhs1, which exhibited a low S-nitrosothiol level and a high seed shattering. However, the background and relevant studies are not well introduced, and the results are presented not in a logic way. This version of manuscript should be improved. Below main concerns I would like the authors to consider while revising the manuscript.
1. In Introduction, lines 83-90, black hull genes is not related with the topic of the manuscript. The sentences could be deleted.
2. Fig.1A, the authors need to explain why two varieties “Woonkwang” and “Drimi-2ho” were used for the comparison with the mutant rhs1. This comparison sounds meaningless. The same issue exists in Fig.4BCD.
3. Fig.1B, if you determine whether RHS1 regulates the expression level of seed shattering related genes (qSH1 and qSH4 and SHAT1) under different SNO conditions, it is better to statistically compare the expression levels between the wild-type and rhs1 mutant.
4. Fig.2B, it is suggested to add some biomass parameters such as seedling length and fresh weight to indicate the sensitivity of the mutants to SNO treatment.
5. Fig. 2, the authors should provide the expression data of OsNOE, because it is associated with NO production and seed-shattering
6. Fig.3, the comparison seems in a wrong way. It should be a pairwise comparison: Dongjin vs. rhs1, Nipponbare vs. noe1. In addition, some information of the gene CPL1 should be provided in the Introduction part, otherwise, it is difficult to follow why you analyze this gene expression.
7. Fig. S4A, the positions of the primers used for mutant identification need to be given.
8. Table S2, significance test values should be added for the gene expression difference.
Author Response
Thanks for the valuable comments. We addressed all the comments diligently.
Please find attached doc

Round 2
Reviewer 1 Report
I am no other questions
Author Response
Thanks for the value comments.
Reviewer 3 Report
I am not satisfied with the revision. The authors do not address my main concerns on the paper. In particular, the concern on the comparison test in Fig. 1b and Fig. 3 is totally ignored, some errors are still there. I fail to see the logic behind the results. Some statements are very confused, for example, lines 288-295, the mutant rhs1 showed high seed shattering but with a low shattering value, while the mutant noe1 showed low shattering with a high shattering value. It is suggested to change the phrase of "shattering value" to "force value" throughout the manuscript.
Author Response
Thanks for the comment. We did not intend to ignore reviewer’s comments, we misunderstood the comment initially. For this round of revision, we have now done the proper comparisons as suggested by the reviewer i.e., Dongjin vs rhs1, Nipponbare vs. noe1. Subsequently, we modified Figure 1 and Figure 3 accordingly.
Furthermore, we have changed the phrase of “shattering value“ to “force value” throughout the manuscript as suggested by the reviewer.
Round 3
Reviewer 3 Report
The quality of this revision is improved. It is acceptable for publication after minor editing. The phrase of "seed pod shattering" is inexact for rice. Please replace "seed pod shattering" with "seed shattering".